# Evidence, Challenges, and Knowledge Gaps Regarding Latent Tuberculosis in Animals

**DOI:** 10.3390/microorganisms10091845

**Published:** 2022-09-15

**Authors:** Pamela Ncube, Bahareh Bagheri, Wynand Johan Goosen, Michele Ann Miller, Samantha Leigh Sampson

**Affiliations:** DSI/NRF Centre of Excellence for Biomedical Tuberculosis Research, South African Medical Research Council Centre for Tuberculosis Research, Department of Biomedical Sciences, Division of Molecular Biology and Human Genetics, Faculty of Medicine and Health Sciences, Stellenbosch University, Francie Van Zijl Dr, Parow, Cape Town 7505, South Africa

**Keywords:** animal tuberculosis, domestic animals, host–pathogen interactions, latent tuberculosis, *Mycobacterium bovis*, *Mycobacterium tuberculosis*, persister bacilli, phenotypic states, wildlife tuberculosis

## Abstract

*Mycobacterium bovis* and other *Mycobacterium tuberculosis* complex (MTBC) pathogens that cause domestic animal and wildlife tuberculosis have received considerably less attention than *M. tuberculosis*, the primary cause of human tuberculosis (TB). Human TB studies have shown that different stages of infection can exist, driven by host–pathogen interactions. This results in the emergence of heterogeneous subpopulations of mycobacteria in different phenotypic states, which range from actively replicating (AR) cells to viable but slowly or non-replicating (VBNR), viable but non-culturable (VBNC), and dormant mycobacteria. The VBNR, VBNC, and dormant subpopulations are believed to underlie latent tuberculosis (LTB) in humans; however, it is unclear if a similar phenomenon could be happening in animals. This review discusses the evidence, challenges, and knowledge gaps regarding LTB in animals, and possible host–pathogen differences in the MTBC strains *M. tuberculosis* and *M. bovis* during infection. We further consider models that might be adapted from human TB research to investigate how the different phenotypic states of bacteria could influence TB stages in animals. In addition, we explore potential host biomarkers and mycobacterial changes in the DosR regulon, transcriptional sigma factors, and resuscitation-promoting factors that may influence the development of LTB.

## 1. Introduction

Tuberculosis is a global threat to animals and humans. In 2020, there were 1.4 million human TB deaths [1]. Although precise numbers are unknown, an estimated 1%–15% of human TB cases are believed to be zoonotic, with transmission originating from animals [2,3,4]. Generally, *M. tuberculosis* infection causes human TB [5], with *M. bovis* the most common infectious agent contributing to animal TB (Table 1) [6]. A wide range of domestic animals (e.g., cattle, goats, sheep, pigs) [7,8,9,10] and wildlife (e.g., buffaloes, elephants, lions, wild dogs, warthogs) [6,11,12] can be infected by *M. bovis* and other *M. tuberculosis* complex (MTBC) species including *Mycobacterium caprae*, *Mycobacterium canettii*, *Mycobacterium pinnipedii*, *Mycobacterium mungi*, *Mycobacterium africanum*, *Mycobacterium suricattae*, *Mycobacterium microti*, *Mycobacterium orygis*, the “chimpanzee bacillus”, and the “dassie bacillus” (Table 1) [13,14,15,16]. Animals with TB can be a source of infection to other animals and humans [17,18]. Given the zoonotic potential and the detrimental effect that TB can have on humans and animals at human–animal–environment interfaces [19], it is important to develop an understanding of TB pathogenesis during different stages of *M. bovis* infection in animals in order to improve control and management strategies.

While there is increased recognition of the complexity of TB stages of infection in humans [20] and animals [21,22], limited studies define the characteristics and diagnosis of these stages in naturally infected animals [21,23,24,25]. The identification of animals that are TB test-positive based on immunological responses [e.g., tuberculin skin test (TST) or interferon-gamma (IFN-γ) release assay (IGRA)] but lack visible lesions on necropsy or are mycobacterial culture-negative, has led to a debate on whether animals can develop LTB [24,25,26,27]. Host-(e.g., genetic variation, immunocompetence, comorbidities) [1,28] and pathogen-specific (e.g., MTBC strains, virulence, infecting dose) [29,30,31] factors likely impact the outcome of infection. However, it is unclear whether these stages in animals mimic those observed in humans, and this requires additional research [32].

One global challenge is that current animal TB control programs in many countries require the culling of any suspected infected animal to prevent its spread [33,34,35]. The culling of TB test-positive animals or depopulation of a herd is usually performed without clear knowledge of the animal’s infection stage; this can lead to unnecessary loss of latently infected animals that are not shedding [36,37,38]. Culling animals has significant negative economic implications for animal owners. Similarly, the removal of wildlife can have negative conservation impacts, including loss of genetic diversity for some high-value species, such as rhinoceros [39,40]. It is also important to consider the implications of latent infection in livestock destined for consumption [18,41,42,43] and wildlife such as buffaloes or lions (which are maintenance hosts) [44,45,46]. In addition, the diagnosis of TB can lead to movement restrictions [40], resulting in financial losses from game sales, hunting, and ecotourism [47]. Therefore, there is a crucial need to improve the understanding of animal TB pathogenesis, infection stages, and epidemiology to inform disease control and management strategies.

Knowledge gaps still exist regarding infection stages in animal TB, as well as host–pathogen interactions and the pathogen metabolic processes involved [48]. Although research on TB diagnosis in different species has progressed [12], studies on TB pathogenesis are complicated by the lack of clinical signs in animals until the disease is advanced, which can take months to years [6,49]. Additionally, TB in wildlife may only be detected while conducting postmortem examinations [50]. Cases with early or subclinical infection can be easily missed without a thorough examination [51], especially during abattoir inspection or field necropsy [52]. These challenges have delayed progress in research on the pathogenesis and elucidation of different infection stages in animals.

The importance of understanding the stages of TB infection in animals stems from the impact it has on the diagnosis and management of infection in individuals as well as intra- and inter-species transmission risk [53]. In turn, this can inform disease control strategies in domestic and wild animals, which affects livelihoods, food security, and conservation [2,39,54]. This review describes the current state of latent TB research in domestic and wild animals and the challenges that still hinder this area of investigation. Specifically, there is a focus on understanding how different hosts (humans vs. domestic animals vs. wildlife) vary in the progression of infection, mycobacterial metabolic activity/pathways that may underly latent TB, and the influence of host–pathogen interactions. Further, we discuss the challenges that exist with TB diagnostic tools in animals. By identifying the knowledge gaps surrounding the progression of TB infection in animals, future investigations can target translational research on animal TB management and control strategies, especially for endangered wildlife and species with high commercial value. This would protect the livelihoods of domestic farmers, especially in developing countries, so as to mitigate TB disease and potential zoonotic outbreaks at human–domestic animal–wildlife interfaces.

## 2. The Epidemiology of TB at the Domestic–Wildlife–Human Interfaces

Targeted epidemiologic studies at domestic–wildlife–human interfaces provide a foundation for effective control, ecological management, surveillance, and eradication of TB [95,146,147,148]. Over the years, the advances in strain-typing methods (such as spoligotyping [149,150,151], variable number of tandem repeats [152,153,154], whole genome sequencing [143,155,156,157,158,159], and other tools for characterising MTBC pathogens [12,22,46,160,161,162] have improved the understanding of genotype distributions [154], demographic risk factors for disease [41,163,164], MTBC geographical distributions of infected animal populations [165,166], and transmission patterns [34,80,167,168]. Transmission and TB manifestations in some hosts are associated with age and sex risk factors [169]. For example, in white-tailed deer, the odds of being TB positive is greater in males than females [95,147,169], highlighting sex-based differences (possibly hormonal) that may influence the risk of TB.

Whole genome sequencing data of MTBC isolates have shown 7 lineage classifications that have different TB epidemiological consequences worldwide [170,171]. Human-adapted MTBC lineages such as *M. africanum* and *M. tuberculosis sensu stricto* are associated with specific geographical locations, variations in virulence, and transmission capacity [172,173,174]. In New Zealand, 782 *M. bovis* genomes (across 8261 locations) over 30 years were used to track TB outbreaks in domestic and wildlife herds [159], demonstrating the significance of whole genome sequencing in understanding infection and transmission patterns. Heterogenous transmission rates and infectivity of *M. bovis* are influenced by community and populations with animal movements playing a key role in TB epidemiology [175,176]. Modifications in animal husbandry practices and changes in ecological and population densities are some factors that can influence transmission events among animals as well as increase zoonotic risks to human populations [177]. For example, it has been shown that modern farming, environmental changes from deforestation, settlement, and expansion of agriculture intensifies TB transmission from animals to humans [177].

The epidemiology of TB in domestic animals such as cattle is well documented in the United States [169], the United Kingdom [178,179,180], Iran [181], Italy [154], France [175], and Spain [182]. However, similar studies in wildlife are limited to some reports from wildlife species such as buffaloes [70], lions [46,88,183], and rhinoceros [93,95,184]. Furthermore, in wildlife settings, the epidemiology and pathogenesis of TB in European badgers have been studied extensively in countries like the United Kingdom and Ireland [46,175]. Badgers are considered maintenance hosts due to their longevity and social structures and are implicated in TB transmission to cattle [147]. Studies have shown that *M. bovis* shed by badgers into the environment can result in transmission to other badgers and cattle (based on genetically related strains) [185,186]. Thus, it is important to fill gaps in epidemiological knowledge on the wide TB host range to improve the detection of infections and mitigate transmission in captive and free-ranging animals.

It has also been demonstrated that domestic animals and wildlife can be an important source of zoonotic pathogens for human populations, particularly in low-income countries [43,187,188,189,190]. Infection with *M. bovis* can result in zoonotic transmission affecting a wide variety of mammals [115,177,191,192]. The zoonotic risk factors include herding infected animals, working in meat abattoirs [108,193], consumption of unpasteurised milk [18,194,195,196], blood [18], and undercooked or raw meat [18,191,193,197]. The zoonotic transmission of tuberculosis from animals to humans and vice versa is an under-reported problem, with few reports on its magnitude, particularly in developing countries [188,198]. However, in recent years, veterinarians, researchers, the World Health Organisation, and the World Organisation for Animal Health have emphasised improving the knowledge base on TB zoonosis using multidisciplinary One Health approaches [41,196,199,200,201,202,203]. This approach should increase understanding of the effects of TB zoonosis on a global scale.

## 3. Response of Different Hosts to MTBC Infection?

### 3.1. Human TB Infection Stages

Characterising different stages of TB infection in humans is an active area of research [204,205], and is typically better understood compared to disease progression in animals [8,120]. A spectrum of clinical stages, ranging from infection clearance, LTB, incipient TB, subclinical TB, and active TB (ATB) disease, has been proposed (Table 2) [32,204]. However, approaches to diagnose, differentiate, and treat these stages are complicated, controversial, and inadequate [20].

Upon infection with pathogenic mycobacteria, there are several potential outcomes. Host defenses may clear the infection; alternatively, the infection may be contained in the form of LTB [146,206], or progress to ATB [204,207]. LTB is defined as an asymptomatic clinical state where individuals are not infectious but may have a risk of progression to active TB disease over time [58]. Characterising the LTB stage in humans is important to prevent disease progression, but it can be difficult to completely eradicate it with existing chemotherapeutic regimens [208]. Therefore, LTB continues to present a global health burden and a major risk factor for the development of ATB disease, particularly in immunocompromised individuals [20,209,210,211].

People with incipient and subclinical TB lack signs and symptoms of disease, while individuals with ATB disease exhibit coughing, night sweats, loss of appetite, weight loss, or yellowish purulent sputum. Symptomatic individuals are a risk to others since they can transmit infectious bacilli in aerosols through speaking, coughing, or sneezing [146,206]. The complex adaptative mechanisms of *M. tuberculosis* in the host [212], host genetic variation influencing TB susceptibility [28,213,214], host-pathogen interactions [171,215,216,217,218], pathogen-pathogen co-infections [105,219], and pathogen-non-tuberculous mycobacterial co-infections [220,221] further complicate the investigation of infection stages. The factors that determine the stages of infection in humans are still unclear but are believed to involve host immune responses and pathogen adaptation mechanisms [222,223].

The hypothesis that different TB infection stages in humans are associated with different mycobacterial metabolic states has been proposed [204,205]. For instance, LTB has been linked to subpopulations of bacilli that are (i) viable but non or slowly replicating (VBNR) [224,225], (ii) viable but non-culturable (VBNC) [226,227], and (iii) metabolically inactive dormant bacilli [228]. Broadly, these subpopulations are categorised in the reversibly drug-tolerant, but not genetically drug-resistant population, termed “persisters” [227,229,230,231]; as shown in (Figure 1). Persister bacteria appear to be associated with LTB, incipient and subclinical infection [204,232,233], while actively replicating (AR) mycobacterial populations are present in ATB [204]. Even though human TB research has progressed, additional research is needed to fully understand these hypothesised TB stages in humans.

### 3.2. Domestic and Wildlife Infection Stages of TB

Domestic animals are considered good models to understand TB pathogenesis and stages due to similar physiology, immunology, and anatomy to humans [30,234,235], and thus may be considered to further understand infection stages in other mammals. Transmission may occur after exposure to infectious secretions including saliva, aerosolised respiratory droplets, urine, faeces, exudate from fistulated lymph nodes, and ingestion of infected tissue by scavengers and predators [49,70,190]. The source of infection may influence the development of clinical signs, which can take months to years to appear, and include nonspecific changes, such as weight loss, reproductive failure, decreased milk production, lymphadenopathy, and coughing [6,8]. However, routes of MTBC infection vary in different host species and this may impact TB pathogenesis and influence the development of clinical signs depending on the number of bacterial loads of the source of infection [236]. Thus, the higher the bacterial load from the source of infection (i.e., ingestion of infected tissue from a TB diseased animal) the higher the likelihood of the infected individual developing clinical signs earlier than a source with a lower bacterial load (i.e., infected grass or water or soil) [18,237,238]. Additionally, signs and sites of infection can vary significantly in different species; for example, elbow hygromas and osteomyelitis in lions [46].

The wide range of animal hosts likely has different immune mechanisms, which would result in variable responses to MTBC infection [31,176,239], adding complications to our understanding of different TB stages. For example, elephants [118,240] and suids [241,242] have an early robust humoral response, whereas carnivores such as lions only develop humoral responses in the presence of disease [88,221,243]. Therefore, when conducting research these immune differences must be considered to improve understanding.

Extrapolation of techniques to diagnose human LTB to the study of natural MTBC infections in different species can be difficult due to the logistics of handling large or dangerous animals, long intervals between infection and development of clinical signs of disease, limited ability to identify the source and route of transmission, as well as a paucity of diagnostic tools [6,8,46,49]. The possibility of latent infections in cattle has been discussed in the literature, however, the evidence is not compelling [24,27,244]. Stages of MTBC infections have been hypothesised based on experimental [21,245] and veterinary field experiences with livestock and wildlife (Table 2). These include cleared infection, LTB, subclinical infection, and ATB disease, based on the presence of viable bacteria, detectable immune responses, clinical signs, and radiographic or gross pathological changes [222]. However, additional concrete evidence for these TB stages in animals is needed, which requires finding solutions to the aforementioned challenges and bridging knowledge gaps through research.

### 3.3. Diagnostic Challenges in Differential Stages of TB

Accurate diagnostic tests are crucial for controlling the spread of TB in humans and domestic, and wild animals [246]. Since animals can be infected with *M. bovis* (or some other MTBC members), definitive diagnosis relies on identifying the specific MTBC organism [247]. One key problem with the microbiological detection of mycobacteria antemortem is the assumption that the individual is shedding at the time of testing [204]. However, this approach is suboptimal, especially when using the limited antemortem samples available in animals, and often leads to false-negative results [74]. Even if MTBC infection is detected, there is a paucity of clinical and diagnostic tools to differentiate between putative LTB, incipient infection or subclinical infection, and ATB disease (Table 3) [248]. In addition, few studies have investigated differences in virulence and other characteristics of MTBC strains that may influence infection stages in animals [249,250].

Latent TB in humans is diagnosed by detectable antigen-specific immunological responses in the absence of clinical symptoms and evidence of pathological changes (Table 3) [4]. The QuantiFERON TB Gold In-tube Plus interferon-gamma release assay is used in human patients that are asymptomatic, lack radiographic changes, and are sputum smear, culture, or PCR negative, to identify LTB [204,206,222]. However, validated mycobacterial antigen-specific IGRAs are only available for a handful of animal species [12], and radiography is limited by the size of the animal as well as logistics, especially with wildlife [28]. For example, presumptive *M. tuberculosis* infection in Asian elephants has been diagnosed using serological responses to specific mycobacterial antigens [240]. However, these individuals may be asymptomatic for years before confirming infection by the culture of trunk wash, tracheobronchial lavage fluids, or post-mortem tissue, and the presence of pathological changes [46].

Since existing TB diagnostic tools can not differentiate between stages of TB infection, any positive test for bovine TB (bTB) usually results in the culling of positive animals, regardless of whether clinical signs or lesions are present [35]. In these cases, a post-mortem inspection may aid in determining if possible latent, subclinical, and active infection is present [241,251]. For example, there are reports of cattle with positive immunological assay results but with no evidence of disease, which could be consistent with LTB [26]. These animals lacked gross and histopathological lesions, yet in vivo TST or in vitro blood-based mycobacterial antigen-specific IGRA was positive, although mycobacterial culture results were negative (Table 3) [26,241,252,253]. Often these animals were considered to have false-positive immunological test results rather than the possibility of having LTB. However, other explanations include suboptimal culture detection in paucibacillary samples or false-positive immunologic test results [26].

In summary, many of the tools available to diagnose LTB in humans are unavailable or have not been thoroughly investigated in animals. There is a lack of validated TB diagnostic tests, which often require species-specific approaches [241]. In animals, respiratory samples are rarely submitted for mycobacterial culture and PCR, due to a paucity of veterinary diagnostic laboratories capable of performing these techniques. Therefore, unless there is a suspicion of TB, the lack of diagnostic workup in animals, especially in wildlife, results in a large knowledge gap to demonstrate the possibility of LTB and other possible infection stages.

## 4. The Influence of MTBC Characteristics on TB Stages

Eleven MTBC ecotypes evolved from *M. canettii* including *M. tuberculosis* and *M. bovis* which are 99.95% genetically similar, and the primary causative agents of TB in humans and animals, respectively [29,249]. MTBC genotyping in different hosts has been studied across Africa including countries like South Africa, Mozambique, Algeria, and Ethiopia with *M. bovis* being the key aetiological pathogen for animal TB [258].

Despite primarily infecting humans, *M. tuberculosis* can occasionally infect domestic animals and wildlife [118,259]. Furthermore, *Mycobacterium bovis* and *M. tuberculosis* share some similar patterns of transmission and pathogenesis [21,25], however, there are genetic [260,261,262], metabolic [263,264], and physiological differences [249,250] between *M. bovis* and *M. tuberculosis* [265], and among *M. bovis* [266] and *M. tuberculosis* [267] strains. These differences could affect granuloma formation [23,268], replicative states during infection, and subsequent infection stages in humans and animals.

*M. tuberculosis* phenotypic heterogeneity has been observed during different in vitro stress conditions such as macrophage exposure and acid stress [214,225,269]. Mouton et al. [225] demonstrated that during macrophage infection, intracellular *M. tuberculosis* showed a relatively homogeneous population structure at early time points, with a similar replication rate at 24 h to the in vitro-cultured bacteria. Yet, from 48 h onwards, a more diverse population emerged, with a distinct slower-growing population at 96 h, demonstrating the ability of *M. tuberculosis* to adapt to stress. However, there are relatively few studies on *M. bovis* phenotypic heterogeneity under in vitro stressors [268]. Therefore, application of in vitro and in vivo stress models, previously utilised for *M. tuberculosis*, to *M. bovis* could reveal new insights into replication dynamics and adaptation to stress.

Similar to *M. tuberculosis* [264,265,269,270,271], in vitro and in vivo studies using *M. bovis* BCG have investigated replication dynamics under different host stressors [117,121,122,123,124]. Results demonstrated how human alveolar macrophages effectively used nitric oxide production as a mechanism to suppress *M. bovis* BCG growth [272], similar to observations with *M. tuberculosis* [273]. In addition, 23 genes likely involved in energy metabolism were downregulated while only a *nar*X gene (encoding nitrate reductase) was upregulated in aerobic dormant BCG [274], suggesting this gene plays a role in dormancy responses, similar to *M. tuberculosis*. Boon and Dick [275] revealed that the dormancy survival regulator (*dos*R) dormancy survival regulator gene that encodes for Rv3133c (α-crystallin small heat shock protein), universal stress protein encoded by Rv2623, and cystathionine β-synthase protein encoded by Rv2626c were upregulated in *M. bovis* BCG during anaerobic conditions. These findings were similar to the behaviour shown by *M. tuberculosis*, in which these genes induced non-replicating or dormant bacilli. Furthermore, the differential abilities of MTBC species to enter a non-replicative state was demonstrated using a rabbit model whereby extrapulmonary tuberculosis was exclusively observed in *M. bovis* Ravenel and *M. bovis* AF2122/97 compared to *M. tuberculosis* CDC1551 and *M. tuberculosis* H37Rv [267].

Adaptation to stressors may provide a greater understanding of differences in the ability of MTBC to become dormant. Exposure of *M. bovis* CRBIP7.106 (isolated from human bone) to cryogenic treatment and nutrient starvation resulted in stress responses including morphological phenotypic adaptation through the formation of rare extreme morphologic forms (L-forms or cell-wall deficient) bacteria [276], likely to be a long-term survival strategy during unfavourable conditions. Similar to *M. tuberculosis*, Rv3134c/devR/devS in *M. bovis* BCG Pasteur strain 1137P2 was responsible for adaptation during nutrient starvation, ex vivo macrophages, and oxygen limitation [277], which would be necessary to create an LTB state. Using a nutrient starvation model, the presence and expression of key genes such as (*mpb83*, *mpb70*, *mmpL8-papA1-pks2* locus, *Mb2651*) that synthesise or encode cell wall, lipid metabolism, and transcriptional regulators, respectively, differed between *M. tuberculosis* H37Rv and *M. bovis* AF2122/97 [260]. For example, a lipid metabolism *mmpL8-papA1-pks2* locus was present only in *M. tuberculosis*. *Mpb83* (encoding cell wall lipoprotein), and *mpb70* (encoding immunogenic lipoprotein) genes were upregulated in *M. bovis* compared to *M. tuberculosis*. In addition, transcriptional regulators showed different patterns with increased expression of *Mb2651*, *Mb2654c*, *Mb3109c/virS*, *Mb3477c* in *M. bovis*, and *Rv0196*, *Rv0275c*, and *Rv2160A-Rv2160c* in *M. tuberculosis*. These observations emphasise the differential way in which the MTBC strains may respond or adapt to stress, thus impacting the ability to result in LTB. Therefore, research into bacterial characteristics that influence host-pathogen interactions and bacterial virulence is essential to elucidate their role in LTB.

Reactivations of *M. tuberculosis* infection in humans are well documented [226,278,279]. Similarly, there are reports of *M. bovis* infection reactivations in elderly people in England [280], immunocompromised elderly from Demark [281] and the Netherlands [163], and in bTB eliminated areas (1% of total cases) [282], suggesting a possible capacity of *M. bovis* to persist in a latent state in humans. In addition, LTB was diagnosed in Mexican abattoir and dairy farm workers infected with *M. bovis;* 76.2% were TST positive and 58.5% IGRA positive [4]. Additionally, cases of BCG infection (BCG-osis) have been reported weeks to months after BCG vaccination in children, suggesting that *M. bovis* BCG can enter a non-replicative persister state as well as reactivate [283,284]. However, more studies are needed to establish *M. bovis* persistence and investigate the relationship between metabolic states of mycobacteria and different infection stages in animals.

## 5. Candidate Models to Improve Understanding of TB Stages in Animals

### 5.1. In Vitro Models

Numerous in vitro experiments have been performed to understand LTB in humans, with a relative paucity of corresponding work using *M. bovis*. In vitro models are pivotal for simulating the microenvironment that mycobacterial pathogens may encounter in human and animal hosts [285,286]. To date, nine variations of hypoxia models have been exploited for *M. tuberculosis* studies [287], while very few have been applied to *M. bovis* strains, although attenuated *M. bovis* BCG strains have been investigated under hypoxic conditions [274]. In one report, *M. bovis* BCG was found to survive and adapt to hypoxia through upregulation of DosR (Rv3133c), an essential dormancy response regulator also found in *M. tuberculosis* [275]. Gideon et al. [288] demonstrated upregulation of genes from the regions of difference (RD) 2 (Rv2659c) and RD11 (Rv2658c) in *M. tuberculosis* during hypoxia [261]. However, RD11 is absent in all *M. bovis* strains, so the hypoxic adaptive advantage this RD region may confer on *M. tuberculosis* would be lost in *M. bovis*. Furthermore, RD2 is absent in *M. bovis* BCG, so this strain may not fully represent the adaptation of *M. bovis* virulent strains [261]. Therefore, it is important to shift research focus from *M. bovis* BCG and commonly applied lab-adapted virulent *M. bovis* Ravenel and *M. bovis* AF2122/97 to pathogenic *M. bovis* strains isolated from animals with natural infections [266]. This approach will build new knowledge on how pathogenic wild-type strains would respond to hypoxia.

Nutrient starvation, acid stress, and multi-stress models that combine single stressors [hypoxia (5% O_2_ content), high CO_2_ (10%), nutrient deprivation (10% Dubos medium), and acidic pH (pH 5.0)] [289] have been explored using *M. tuberculosis* to mimic multiple host-pathogen interactions upon infection [289]. However, the use of these models with *M. bovis* has been limited. In vitro nutrient starvation of *M. bovis* BCG induced upregulation of the *Rv3134c*/*devR*/*dev*S transcriptional regulator, which is also present in *M. tuberculosis* [277]. Further, *M. bovis* BCG was found to be tolerant of low pH through the adoption of a homeostatic balance between the internal and external pH [290]. Another study found that *M. bovis* BCG was also tolerant to low pH in the presence of glutamate as a carbon source [291]. These results highlight that in an acidic environment, a subpopulation of *M. bovis* bacilli, which can utilise glutamate as a carbon source, may have the ability to persist longer than those that cannot utilise glutamate. Thus, these in vitro models appear to be a potential platform for additional investigations using field strains of *M. bovis*.

In vitro macrophage infection models have been widely applied to understand differences in pathogenesis and host tropism in *M. bovis*, *M. tuberculosis* [265], *Mycobacterium avium subsp. paratuberculosis*, and *Mycobacterium avium subsp. avium* [292]. Bovine alveolar macrophages [265], blood-derived macrophages [293], human monocyte THP-1 [294], and RAW 264.7 macrophages [295,296] have been used to understand mechanisms that *M. bovis* employs to adapt to stress, host-pathogen interactions, and vaccine discovery. Bovine alveolar macrophages infected with *M. bovis* BCG and two virulent *M. bovis* strains (*M. bovis* Ravenel and *M. bovis* AF2122/97) have revealed transcriptional and proteomic differences between *M. bovis* and *M. tuberculosis* pathogens under acid shock, hypoxia, and macrophage infection [260,297,298,299]. These studies showed that *M. tuberculosis* expressed higher levels of nitrate reductase-associated genes *narG* and *narH*, while *M. bovis* expressed more *mpb70* and *mpb83* (the two most upregulated genes of the overall analysis) [260,298]. A cell wall-associated protein, DipZ (member of sigK regulon), that encodes for MPB70 and MPB83, was among the top ten highly expressed proteins in *M. bovis* compared to *M. tuberculosis* under stress conditions. Additionally, Rv0188, an enduring hypoxic response antigen, was found to be solely associated with low pathology scores in cattle infected with *M. bovis*, suggesting that it may play a role in inducing a non-replicative state [300]. These genes and protein expression differences could potentially influence TB stages in humans and animals [27]; however, additional studies are needed to investigate their effects on the development of LTB using diverse hosts [28,301,302] and MTBC strains. Since bovine alveolar macrophages are isolated from cattle, this model may be considered a more physiologically relevant model for *M. bovis* and thus suitable for understanding the host-pathogen interactions during *M. bovis* infections.

### 5.2. In Vivo Models

Cattle are the most commonly used animals for investigating in vivo *M. bovis* pathogenesis and host-pathogen interactions [25,300,303]. Although cattle are a natural host of *M. bovis*, they may not recapitulate infection stages in the wide variety of other animal species infected with *M. bovis* [268,304]. Experiments to study LTB using cattle have a logistical challenge since there are limited antemortem methods to confirm whether pathological changes are present or absent [244]. Cattle experimentally infected with *M. bovis* often develop clinical signs and progress to disease [268], whereas infections with *M. tuberculosis* H37Rv or other *M. tuberculosis* isolates are reported to delay progression to disease [31,120,244]. Therefore, *M. bovis* dose-response and strain differences in experimental infection models need further exploration.

Domestic pigs have been recently used as a model to understand domestic animals and human TB [30,305,306,307]. Pigs experimentally infected with *M. bovis* AF2122/97 exhibited greater morbidity, rapid onset mortality, severe pathology, and weight loss, as compared to less severe disease in those infected with *M. tuberculosis* Erdman [30]. *M. tuberculosis*-challenged pigs developed the active disease but had delayed onset of fever, no signs of respiratory distress, and showed no change in health status, compared to *M. bovis* infection, during the 35-week trial [30]. Therefore, the pig model may be useful for exploring whether different *M. bovis* field strains develop different infection stages compared to the commonly used *M. bovis* AF2122/97 and *M. bovis* Ravenel strains.

The badger is a promising in vivo model for studying LTB in animals [21,308,309]. This model mimics observations in naturally *M. bovis*-infected animals, with up to 80% of infected badgers containing *M. bovis* for years, until reactivation occurs when their immune system becomes compromised [308,309]. Chronic progressive pulmonary TB occurs in badgers, therefore this model may be a candidate system for not only LTB but also for subclinical infection studies [308].

The Cornell mouse model is considered a “classic” TB latency model, in which infected mice are treated with antibiotics to reduce bacteria to undetectable levels, before TB reactivation [279,310]. This model results in a steady asymptomatic state that resembles human latency in which bacteria enter a slowly or non-replicating state [205,311,312,313]. A study found that mice vaccinated before being challenged developed different immunopathology, with *M. bovis* BCG challenged mice having lower histopathological scores than those challenged with *M. tuberculosis* [314]. Although this is not surprising since BCG is attenuated, it raises the question of whether virulent *M. bovis* strains would differ in their ability to enter a slowly or non-replicating state in this model.

The guinea pig TB model has also been used to study infection stages [315], *M. bovis* diagnostics [316], and vaccine development [317]. This has been used to study LTB using *M. tuberculosis* [318] but hasn’t been evaluated using *M. bovis* clinical strains. Mice and guinea pig models are relatively cheaper than those using large animals, such as cattle and pigs, although they can be technically challenging and present limitations such as lack of granuloma structure and organisation, no extracellular dissemination, and difficulty establishing LTB without vaccinations [222,317,319] Therefore, badgers, cattle, and pigs may be more physiologically relevant to studying the different TB stages in *M. bovis* natural hosts.

*In vivo* research is needed to answer questions about the existence of different TB stages in animals. Although cattle, pig, and badger models appear to be the most appropriate models, they do not produce all features of the different TB stages or reflect the diverse host-pathogen repertoire of responses during infection in other domestic animal and wildlife species [314,315,320].

## 6. Candidate Host and Pathogen Biomarkers to Improve Understanding of TB Stages

### 6.1. Candidate Host Markers

Currently, there are no specific host immunologic biomarkers definitively associated with latency in humans and animals because most overlap with those for active disease [48,223,285,321], although in vitro and in vivo models have shed light on possible candidates [205]. A quantitative comparison of cytokine levels in the blood of latently infected versus actively diseased patients showed differences in tumor necrosis factor α, IFN-γ, and interleukin 12 (IL-12) [26,48,205]. Transcriptomic profiling of immune responses in a Gambian study found that an antiapoptotic gene *bcl2* was a promising marker that indicated the onset of active disease very early after infection [322]. In addition, the study found that the B-cell lymphoma 2 (BCL2) marker was significantly lower in individuals with ATB disease versus latently infected individuals [322], also making it a potential host candidate biomarker. Another study found that IL-13 and autoimmune regulator (AIRE) were biomarkers that could identify high-risk people with LTB that develop ATB, months before clinical diagnosis [323].

Multiple cytokine assays have been evaluated in human studies for their ability to differentiate active disease from LTB [324,325]. Suzukawa et al. [326] found that combined analysis of IFN-γ, IL-2, IL-5, IL-10, IL-1RA, and MCP-1/CCL2 cytokines found in QuantiFERON supernatant, could distinguish ATB from LTB in humans. In a recent study, in vitro antigen-specific concentrations of IFN-γ, IL-2, IL-5, IL-10, IL-1RA, and monocyte chemoattractant protein-1 (MCP-1/CCL2) showed promise for distinguishing ATB from LTB [321,326]. A recent systematic review that compared human host factors involved in LTB, identified IL-1, IL-6, IL-9, IL-17, and IL-IRA as potential candidates for identifying human latency [48].

In contrast, there are relatively few studies on diagnostic host biomarkers in animal TB, with the majority focusing on developing diagnostic tools and investigating pathogenesis [12,21,31,252]. Therefore, additional investigation is needed to confirm whether these can be used to detect latent infection in animals. The development of an LTB cytokine panel would need to consider the variability of MTBC characteristics, including bacterial heterogeneity, host immunological responses, and pathogenesis, across animal species.

### 6.2. Candidate Pathogen Biomarkers

Several mycobacterial factors are upregulated during *M. tuberculosis* infection in response to host defences. In human LTB, these include genes that mediate energy metabolism, lipid production, stress response, growth restriction, hypoxia response, and toxin-antitoxin stress gene regulators of the bacilli [205,227,327]. By extrapolating from human LTB to animals, candidate bacterial biomarkers to investigate would be DosR and other transcriptional factors.

The DosR regulon is one of the best-studied regulatory systems in mycobacterial responses to anaerobic conditions [275,328]. DosR is responsible for the maintenance and upkeep of mycobacterial persister populations believed to underlie latency and is conserved in *M. tuberculosis* [227,329,330,331] and *M. bovis* [277]. This regulon is a 48-component system controlled by the DosR regulator [270]. *Dos*R genes include *nrdZ*, *narX*, *narK2*, *ctpF*, *otsB1*, *fdxA*, *pdxA*, *pfkB*, *acr*, *acg*, and *dosR*, which respond to environmental stress conditions, leading to mycobacterial dormancy and persistence [270,332,333,334]. The DosR antigen Rv2031c (Acr or HspX) has been associated with adaptation to stressful in vitro conditions such as hypoxia [297,335]. Importantly, studies in cattle infected with *M. tuberculosis* H37Rv and *M. bovis* AF2122/97 [244] have demonstrated that this antigen may differentiate between active disease and possible LTB in animals.

A recent study has recognised the potential of DosR and resuscitation-promoting factors (Rpf) antigens [Rv2029c (PfkB), Rv2389c (RpfA), Rv0867c (RpfA), and E6-C1] to discriminate between LTB and ATB in a TB endemic human population [330]. Similar results have been observed in other studies conducted in African, Asian, European, Indian, and Brazilian populations [330,331,336], suggesting that DosR antigens are important for latency, independent of host immune responses, environmental background, human genetics, and circulating *M. tuberculosis* strains. Furthermore, a study by Jones et al. [300] identifies potential LTB markers in naturally infected cattle such as DosR/S/T antigens (Rv2627c, Rv2628, Rv2029c, and Rv1733c). However, the number of animals with no visible lesions was very low, so LTB may have been missed due to the experimental conditions. Therefore, studies with a larger sample size would provide further clarity and a better understanding of LTB biomarkers.

The main driver for latent *M. tuberculosis* infection is believed to be hypoxia [288], but more recent studies have discovered additional factors that could play a role. Transcription of mRNA sigma factor genes is implicated in different stress responses and may play a role in the adaptation to stressful host environments [337]. These include sigma factor F gene (*sigF*) (responsible for slowing growth of bacteria) [338,339], and *hspX/ acr*/ *Rv2031c* genes that encode for the α-crystallin homolog, and small heat shock protein (sHSP16.3) [340,341]. *SigF* is upregulated in *M. tuberculosis* persisters, suggesting a role in latency [332]. Further, the *sigF* gene is responsible for regulating oxidative stress, antibiotic stress, cold shock, and nutrient depletion [332,337]. Sigma factor B (*sigB*) and *sig**H* are additional genes induced by heat and oxidative stress [342,343]. Although *sigF* has been demonstrated to be induced in hypoxic conditions and the stationary growth phase in *M. bovis* BCG [337,344], there is a need to further investigate transcriptional factors specifically involved in *M. bovis* persister bacilli.

The tools and methods to study *M. tuberculosis* persistence have improved in recent years, hence some mycobacterial candidate genes for persistence may overlap in *M. tuberculosis* and *M. bovis* strains due to their genetic similarities. These similar genetic characteristics could make it easier to reproduce models using *M. bovis* or other MTBC species, thus accelerating latency research in animals. Nevertheless, some of the physiological differences between these strains may reveal unique candidate genes for latency upon infection and this must be taken into consideration during experimental design.

## 7. How Can Researchers Bridge Existing LTB Knowledge Gaps in Animals?

Although laboratory-based research has highlighted the divergent nature of *M. tuberculosis* and *M. bovis* infection stages in mice [314], guinea pigs [290], cattle [31], badgers [21], and domestic pigs [307,345], there is still no compelling evidence on whether LTB occurs in domestic and wild animals infected with *M. bovis*, *M. tuberculosis*, or other MTBC pathogens. Therefore, there is a need to identify knowledge gaps and possible approaches to address these.

### 7.1. Factors Contributing to Existing Knowledge Gaps

A paucity of studies describing and comparing the pathogenesis of *M. bovis* infection and other MTBC species in different animal species.Limited availability of sensitive and specific diagnostic tools for detection and differentiation of MTBC infection stages in domestic animals and wildlife, especially antemortem tests.Incomplete information on the diversity of MTBC virulent clinical strains, primarily *M. bovis*, and their influence on pathogenesis.A limited understanding of the role and variability in immune responses to mycobacterial infection in different animal species.A poor understanding of the genetic, metabolic, and physiological characteristics of *M. bovis* could promote persister bacilli formation.A lack of clarity on how TB stages vary in different hosts (humans, domestic and wild animals).A lack of well-characterised in vitro and in vivo models of *M. bovis* infection to simulate different stages of infection, including LTB.

### 7.2. Recommendations for Future Research

To improve understanding of LTB in animals, data from in vitro and in vivo MTBC archetypal strains (such as *M. tuberculosis* H37Rv, *M. tuberculosis* CDC1551, *M. bovis* Ravenel, and *M. bovis* AF2122/97) and studies on host responses in *M. tuberculosis* and *M. bovis* BCG infections should be used to design future research. This should include:Developing a consensus on the definition of latency in domestic and wild animals and identifying a model that could be used to find biomarkers for this state.Identification of blood-based host and pathogen biomarkers that can differentiate between ATB and different stages of *M. bovis* infection in different animal species.Utilising available tools to study the phenotypic state of persister bacilli at a single-cell level to understand the physiological, phenotypic, and molecular features of different strains.Comparing the pathogenesis of *M. bovis* and other MTBC in different animal species to characterise the chronic asymptomatic state in infected hosts.Exploring host–pathogen similarities and differences of host–pathogen interactions to elucidate factors leading to LTB and susceptibility of different species to latent infections.

## 8. Conclusions

Animal TB research is still lagging compared to human TB research. Reasons include the complexity introduced by the fact that animals can be infected by a wide range of MTBC species, and a range of animal species can be infected by *M. bovis* [176]. Further, each MTBC species has diverse strains which could have differential influences on their pathogenesis and virulence. There is a lack of compelling evidence to prove the existence of different TB stages in animals as proposed in humans [204]. Experimental cattle [31] and pig studies [30], that applied laboratory-adapted strains, suggest that *M. bovis* is less likely to enter a persister state than *M. tuberculosis*. Comparative genetic and proteomic studies of *M. bovis* and *M. tuberculosis* have shown metabolic and replicative differences which could impact how these pathogens adapt to different animal hosts [249,250]. Despite these genetic and proteomic differences, reports of *M. bovis* reactivation in humans suggest that this species is capable of establishing LTB. However, further investigations are needed to ascertain the likelihood of MTBC pathogens (especially *M. bovis*) establishing LTB in animals and humans [250,267].

Although there is a lack of compelling evidence for latency in cattle [27,160], some ante-mortem and post-mortem findings suggest a possible LTB stage in animals [21,26,131,308]. Latent TB could occur in some animal species but is under-recognised due to the lack of defined criteria for TB stages, the challenging logistics, and the lack of diagnostic tools that can differentiate between several infection stages. However, there are promising approaches to studying human LTB that could serve as a foundation to elucidate this stage in animals [204]. Blood-based test approaches present a convenient way to distinguish TB stages ante-mortem; however, the current biomarkers of LTB and active disease overlap [326]. While post-mortem tests can also be applied to validate LTB in animals they can only be performed upon culling an individual, therefore presenting feasibility and cost implications. These myriad challenges negatively impact the progress of research to determine if LTB occurs in domestic or wild animals, however, using cross-species studies (*M. bovis* vs. *M. tuberculosis*) could advance understanding of LTB at the domestic-wildlife-human interfaces.

Identifying TB stages such as LTB in animals can inform policy, devise targeted management, and improve control strategies based on associated risks. Studying LTB in animals could also differentiate individuals that may be infected but not infectious from those with a greater risk of spreading the disease, provide potential management changes that could prevent infected animals from progressing to active disease (and therefore spread) and minimise the loss of animals due to unnecessary culling (especially for endangered animals such as rhinos). In vitro and in vivo models can be applied to broaden knowledge on whether different field MTBC species, particularly *M. bovis*, induce persister bacilli believed to underlie LTB. When designing such studies, it is important to be aware of the diversity in host-pathogen interactions and infection outcomes in different animals. Moreover, the knowledge gained from animal TB research can also inform models to study LTB in humans. Future research should also focus on methods to further identify persister bacterial populations, not only in culture but also directly in animal samples to characterize their phenotypes, ultimately bridging some LTB knowledge gaps.

## Figures and Tables

**Figure 1 microorganisms-10-01845-f001:**
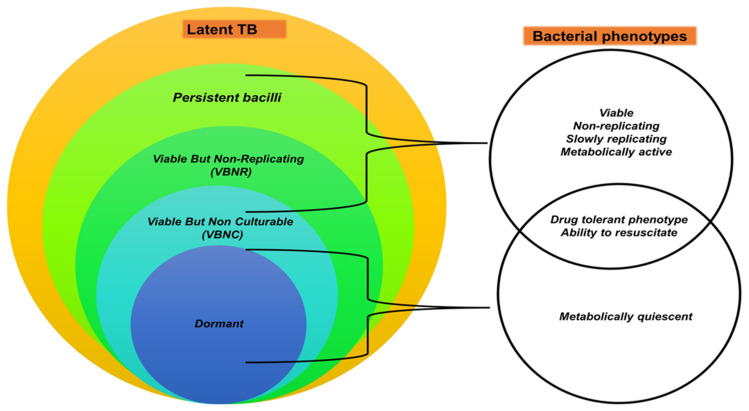
Latent TB stage and underlying bacterial phenotypes. Latent TB is speculated to encompass several bacterial phenotypes, under the broad heading of persister bacteria. The subpopulations of persister bacilli include viable but non-replicating (VBNR) bacilli, viable but non-culturable VBNC bacilli, and dormant bacilli. These have varying phenotypic characteristics and may vary with each TB stage. (Original figure).

**Table 1 microorganisms-10-01845-t001:** *Mycobacterium tuberculosis* complex pathogens and their hosts.

MTBC Pathogens	Primary Hosts	Secondary/Sporadic Hosts
*M. bovis*	African buffaloes (*Syncerus caffer*) [44,55,56], Asian elephants (*Elephas maximus*) [57,58,59,60,61,62], bushbucks (*Tragelaphus scriptus*) [28,63,64], bush pigs [40,65,66] (*Potamochoerus porcus*), cattle [18,67,68,69], chacma baboons (*Papio ursinus*) [28,70,71], cheetahs (*Acinonyx jubatus*) [70,72], common duiker (*Syvicapra grimmia*) [12,73], African elephants (*Loxodonta africana*) [28,59,74], eland (*Taurotragus oryx*) [28,40,66], giraffe (*Giraffa camelopardalis*) [75], european badger (*Meles meles*) [76,77,78,79], honey badger (*Mellivora capensis*) [12,28,80], greater kudu (*Tragelaphus strepsiceros*) [72,81], hippopotamus (*Hippopotamus amphibius*) [82,83], impala (*Aepyceros melampus*) [28,40,84], African leopards (Panthera pardus) [28,35,85], African lions (*Panthera leo*) [6,11,28,86,87,88,89], banded mongoose (*Mungos mungo*) [90,91], nyala (*Tragelaphus angasii*) [28,92] black rhinoceros (*Diceros bicornis*) [6,93,94,95], red fox (*Vulpes vulpes*) [96], spotted genet (*Genetta tigrine*) [28,65,81], springbok (*Antidorcas marsupialis*) [12,28], common warthog (*Phacochoerus africanus*) [28,97,98,99], wild boars (*Sus scrofa*) [96], African wild dog (*Lycaon pictus*) [100,101,102,103], blue wildebeest (*Connochaetes taurinus*) [104,105], wild deer (*Capreolus capreolus*, *Cervus elaphus*) [96], white rhinoceros (*Ceratotherium simum*) [35,95,104,106]	Humans [17,19,43], cats [107]
*M. tuberculosis*	Humans [1,19,28]	Cattle [108,109], non-human primates [110,111], dogs [112,113,114], cats [115], zoo animals e.g., black and white rhinos [116], African elephants, and Asian elephants [117,118,119], goats [120]
*Mycobacterium africanum*	Humans [121,122,123]	Cattle [124], rock hyraxes (*Procavia capensis*) [29,125]
*Mycobacterium pinnipedii*	Seals (*pinnipedts*) [126]	Humans [127,128], cattle [129]
*Mycobacterium caprae*	Goats, sheep, and pigs [130,131]	Humans [132]
*Mycobacterium microti*	Cats, pigs [133], bank vole, wood mouse, shrew [134]	Humans [135]
*Mycobacterium orygis*	Antelope, deer, waterbuck [136], oryxes, gazelles [137],	Humans [138]
*Mycobacterium canettii*	Humans [29,139,140,141]	-
*Mycobacterium mungi*	Banded mongooses [142]	-
*Mycobacterium suricattae*	Meerkats (*Suricata suricatta*) [143]	-
*Chimpanzee bacillus*	Chimpanzee (*Pan troglodytes*) [16]	-
*Dassie bacillus*	Rock hyraxes [14,144], meerkats [143,145]	-

**Table 2 microorganisms-10-01845-t002:** Hypothetical stages of mycobacterial infection and host–pathogen interactions in domestic and wild animals. Infection stages may vary in different species depending on the host’s immune response and risk factors, as well as strain and infecting dose of *Mycobacterium bovis*. (Adapted and adjusted from [25,204]).

bTB Stage		Clinical Signs	Host–Pathogen Interactions
**Uninfected**		Absent	- No immune response, and no viable bacteria
**Infected**	Cleared infection	Absent	- Innate and/or adaptive immune system clears the bacteria - There may be no memory response, or the immune memory response may last for some time after the pathogen has been eliminated - No viable bacteria present
	Latent infection	Absent	- Detectable immune responses present - Viable and slow or non-replicating bacteria possibly present
	Subclinical infection	Absent	- A transition state from latency or initial/early infection to active disease - Viable and slow or non-replicating bacteria likely present along with a small proportion of actively replicating populations - Microbiological evidence of culturable bacteria - Radiographic, gross, or histopathological evidence of infection
	Active TB disease	Present	- Viable and replicating bacteria present - Clinical signs (e.g., coughing, weight loss, and lymphadenopathy) - The individual may be infectious (intermittent shedding) - It is relatively easy to find culturable bacteria depending on the location of infection and the type of samples used for analysis

**Table 3 microorganisms-10-01845-t003:** Hypothetical infection stages and associated antemortem and postmortem test results for categorising TB stages.

Animal TB Stage (a)		Antemortem Tests (b,c)		Postmortem Tests (d)	
		TST	IGRA	Bacterial Culture	Tissue Histopathology Consistent with bTB (Yes/No)
**Uninfected**		Negative	Negative	Negative	No
**Infected**	Cleared infection	Negative (innate immune system clearance) Positive (adaptive system immune clearance)	Positive for a short period before waning (adaptive immune clearance)	Negative	No
	Latent infection	Positive	Positive	Negative	No
	Subclinical infection	Positive	Positive	Positive	Yes (early-stage granulomas)
	Active disease	Positive	Positive	Positive	Yes (multifocal and/or confluent late-stage granulomas

TST = tuberculin skin test, IGRA = interferon-γ release assay. References: **a** [204], **b** [26,254,255], **c** [256,257], **d** [24,25,131,222].

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
