# Peer review of "Evidence, Challenges, and Knowledge Gaps Regarding Latent Tuberculosis in Animals"

_microorganisms, 2022, doi:10.3390/microorganisms10091845_

Round 1

Reviewer 1 Report

This is an interesting paper summarizing the knowledge on MTC infections in animals pointing an important natural reservoir for tuberculosis. The zoonotic transmission of tuberculosis is a raising issue and it should be pointed out. The paper is well written and all the necessary references have been included.  A table with all the MTC species and their hosts would be nice to be included to facilitate the reader and to remind the basics. Also a summary of any epidemiological studies in animals if available should be added. 

Reviewer 2 Report

Title:  Evidence, challenges, and knowledge gaps regarding latent tuberculosis in animals

General Comment: The manuscript discusses host-pathogen interactions between the MTBC strains M. tuberculosis and M. bovis during infection as well as the evidence, difficulties, and knowledge gaps surrounding LTB in animals. To further explore how the various phenotypic states of bacteria might affect the phases of TB in animals, the authors have taken into consideration models that could be modified from research on human tuberculosis. The authors also investigate transcriptional sigma factors, resuscitation-promoting factors, and mycobacterial alterations in the DosR regulon that may affect the development of LTB.

My comments are the following:

Line 12            Italicize Mycobacterium bovis and Mycobacterium tuberculosis through out the manuscript.

Table 1            "There be no memory response," should be "There may  be no memory response,"

Page6 Line2     Please correct the heading

Page6 line 38  ‘defferential’ should be ‘differential’

Page 51           Italicize the gene names

Line95             Please correct O2 and CO2 to O2 and CO2

The authors can discuss the nematode,  C. elegans  as an in vivo model to study Mycobacterial transmission. An example reference is "Establishment of a Host-to-Host Transmission Model for Mycobacterium avium subsp. hominissuis Using Caenorhabditis elegans and Identification of Colonization-Associated Genes"
